# Role of PB1 Midbody Remnant Creating Tethered Polar Bodies during Meiosis II

**DOI:** 10.3390/genes11121394

**Published:** 2020-11-24

**Authors:** Alex McDougall, Celine Hebras, Gerard Pruliere, David Burgess, Vlad Costache, Remi Dumollard, Janet Chenevert

**Affiliations:** 1Laboratoire de Biologie du Développement de Villefranche-sur-Mer (LBDV), Institut de la Mer (IMEV), Sorbonne Université/CNRS, 06230 Villefranche sur-Mer, France; hebras@obs-vlfr.fr (C.H.); pruliere@obs-vlfr.fr (G.P.); costache@obs-vlfr.fr (V.C.); dumollard@obs-vlfr.fr (R.D.); chenevert@obs-vlfr.fr (J.C.); 2Biology Department, Boston College, 528 Higgins Hall, 140 Commonwealth Ave, Chestnut Hill, MA 0246, USA; david.burgess@bc.edu

**Keywords:** midbody remnant, second polar body, meiotic spindle, ascidian

## Abstract

Polar body (PB) formation is an extreme form of unequal cell division that occurs in oocytes due to the eccentric position of the small meiotic spindle near the oocyte cortex. Prior to PB formation, a chromatin-centered process causes the cortex overlying the meiotic chromosomes to become polarized. This polarized cortical subdomain marks the site where a cortical protrusion or outpocket forms at the oocyte surface creating the future PBs. Using ascidians, we observed that PB1 becomes tethered to the fertilized egg via PB2, indicating that the site of PB1 cytokinesis directed the precise site for PB2 emission. We therefore studied whether the midbody remnant left behind following PB1 emission was involved, together with the egg chromatin, in defining the precise cortical site for PB2 emission. During outpocketing of PB2 in ascidians, we discovered that a small structure around 1 µm in diameter protruded from the cortical outpocket that will form the future PB2, which we define as the “polar corps”. As emission of PB2 progressed, this small polar corps became localized between PB2 and PB1 and appeared to link PB2 to PB1. We tested the hypothesis that this small polar corps on the surface of the forming PB2 outpocket was the midbody remnant from the previous round of PB1 cytokinesis. We had previously discovered that Plk1::Ven labeled midbody remnants in ascidian embryos. We therefore used Plk1::Ven to follow the dynamics of the PB1 midbody remnant during meiosis II. Plk1::Ven strongly labeled the small polar corps that formed on the surface of the cortical outpocket that created PB2. Following emission of PB2, this polar corps was rich in Plk1::Ven and linked PB2 to PB1. By labelling actin (with TRITC-Phalloidin) we also demonstrated that actin accumulates at the midbody remnant and also forms a cortical cap around the midbody remnant in meiosis II that prefigured the precise site of cortical outpocketing during PB2 emission. Phalloidin staining of actin and immunolabelling of anti-phospho aPKC during meiosis II in fertilized eggs that had PB1 removed suggested that the midbody remnant remained within the fertilized egg following emission of PB1. Dynamic imaging of microtubules labelled with Ens::3GFP, MAP7::GFP or EB3::3GFP showed that one pole of the second meiotic spindle was located near the midbody remnant while the other pole rotated away from the cortex during outpocketing. Finally, we report that failure of the second meiotic spindle to rotate can lead to the formation of two cortical outpockets at anaphase II, one above each set of chromatids. It is not known whether the midbody remnant of PB1 is involved in directing the precise location of PB2 since our data are correlative in ascidians. However, a review of the literature indicates that PB1 is tethered to the egg surface via PB2 in several species including members of the cnidarians, lophotrochozoa and echinoids, suggesting that the midbody remnant formed during PB1 emission may be involved in directing the precise site of PB2 emission throughout the invertebrates.

## 1. Introduction

Polar body (PB) emission occurs in oocytes/fertilized eggs and is an extreme form of unequal cell division. One defining feature of meiosis is that two successive rounds of M phase and cytokinesis occur without an intervening S phase. At the end of the first meiotic M phase (meiosis I) in animal oocytes, cytokinesis creates PB1 which is immediately followed by the second meiotic M phase (meiosis II) and a second round of cytokinesis to produce PB2. This generates one oocyte (or zygote) and two polar bodies (although three PBs are sometimes observed since PB1 can divide). In chordates, this extreme form of unequal cell division depends on the actin-dependent migration of the first meiotic spindle from the oocyte interior to the cortex [1,2,3]. Once at the cortex, the chromosomes cause a small subdomain of the overlying cortex to become polarized [4,5,6]. More recent findings in mouse oocytes have demonstrated that a chromatin-centered Ran-GTP gradient polarizes a cortical subdomain in close proximity to the meiotic chromosomes [7]. In mouse oocytes, the chromatin-centered Ran-GTP gradient was observed using a Fret-based biosensor [8]. These findings were prompted by earlier work showing that spindle assembly can be caused by a chromatin-centered Ran-GTP gradient induced by chromatin localized RCC1 [9,10,11]. One essential finding therefore is that a chromatin-centered Ran-GTP gradient causes a subdomain of the cortex adjacent to the meiotic chromosomes to become polarized driving PB outpocketing [7,12].

PB formation is a stepwise process beginning with cortical polarization, followed by protrusive outpocketing of the polarized cortex and ending with constriction of the cortical outpocket. The protrusive outpocket therefore defines the bulge which will form the future polar body. During meiosis I in mouse oocytes, a chromatin-centered Ran-GTP gradient promotes the formation of an actin cap via the inactivation of ERM (Ezrin/Radixin/Moesin) independent of Cdc42 [12]. Cortical outpocketing is initiated at the actin cap during anaphase and is accompanied by the creation of dynamic actin via the recruitment of Cdc42, N-WASP and Arp2/3 in mouse [12] and *Xenopus* oocytes [13,14]. Cdc42 is required for cortical outpocketing in mouse oocytes, since the dominant negative Cdc42 or specific deletion of Cdc42 prevents cortical outpocketing [15,16]. It is not entirely known how the fall in MPF activity triggers the cortical recruitment of active Cdc42 during anaphase, which can be abolished by preventing the fall in MPF activity [14]. Outpocketing also occurs during meiosis II, and similarly to PB1 emission, in mouse oocytes the active form of Cdc42 becomes enriched at a cortical subdomain adjacent to one spindle pole and chromatids that are closest to the cortex [15]. During PB cytokinesis, ECT2 localized at the spindle midzone leads to the formation of a RhoA contractile ring rich in myosin II [17]. Thus, cortical outpocketing during anaphase I and II is thought to be induced by the cortical accumulation of active Cdc42 in *Xenopus* and mouse oocytes. However, it is not entirely clear what triggers the accumulation of active Cdc42 at the cortex driving outpocketing [18], and indeed how one outpocket rather than two are formed during meiosis II. Here in the ascidian we found that during meiosis II the midbody formed during PB1 emission becomes visible as a small “polar corps” sitting on the protrusive outpocket and that this polar corps predicts the precise site of PB2 outpocketing; we thus speculate that the polar corps may be involved in attracting one spindle pole into the protrusive outpocket during Ana II. We speculate that this may result in PB2 being emitted at the precise site of PB1 cytokinesis, thereby resulting in PB1 becoming tethered to the egg indirectly via PB2. Such tethered polar bodies appears to be a widespread occurrence throughout invertebrate species including cnidarians, lophotrochozoa, and echinoderms.

## 2. Materials and Methods

### 2.1. Origin of the Animals

Adult animals of *Phallusia mammillata* and *Mytilus galloprovincialis* were collected at Sète (Etang de Tau, Mediterranean coast, France). Ascidian gamete collection, dechorionation, fertilization and embryo cultures were as described previously [19]. For example, to dechorionate ascidian eggs chorionated eggs were treated with 1% trypsin solution for 90 min, gently pipetted and the denuded eggs free of their extracellular chorion were washed three times to replace the trypsin seawater with fresh seawater.

### 2.2. Microinjection, Imaging and Reagents

Microinjection was performed by inserting about 50 eggs into a holding chamber (wedge) made from glass pieces stuck to a 22 mm coverslip with valab (Vaseline:lanolin:beeswax 1:1:1). Dechorionated eggs were mounted in glass wedges and injected with mRNA (1–2 µg/µL pipette concentration/ ~1–2% injection volume) using a high pressure system (Narishige IM300, London, UK). mRNA-injected eggs were left for 2–5 h or overnight before fertilization and imaging of fluorescent fusion protein constructs. Epifluorescence imaging was performed with an Olympus IX70, Zeiss Axiovert 100 or Axiovert 200 equipped with cooled CCD cameras and controlled with MetaMorph software package. Confocal microscopy was performed using a Leica SP5 or SP8 fitted with 40x/1.3 na oil objective lens and 40x/1.1 na water objective lens. All live imaging experiments were performed at 18–19 °C.

Fixation and labelling for immunofluorescence. Eggs were fixed in −20° methanol containing 50 mM EGTA, blocked with PBS containing 2% BSA, and incubated with anti-tubulin primary antibody DM1a (Sigma-Aldrich, Lyon France) at a dilution of 1:500 and TRITC-conjugated anti-mouse secondary antibody (Santa Cruz, Lyon, France) at a dilution of 1:200, washed in PBS and mounted in Citifluor (Biovalley, Thermo Fisher, Illkirch, France, AF1-100). Using the same fixation procedure, fixed eggs were incubated with anti-phospho aPKC antibody (Santa Cruz, sc-12894-R) at a dilution of 1:100 and FITC-conjugated anti-rabbit secondary antibody (Santa Cruz) at a dilution of 1:200. For TRITC-Phalloidin (Molecular Probes, Thermo Fisher, Illkirch, France R415), activated eggs were fixed in 3.7% formaldehyde in 0.5 M NaCl in PBS 10 and 12 min. after PB1 emission. After several washes in PBT (PBS containing 3% BSA and 0.05% Triton X-100), the eggs were stained using TRITC-Phalloidin (10 µg/mL) and Hoechst 33342 (1 µg/mL). LifeAct::GFP or LifeAct::mCherry protein was made in bacteria, purified (8 µg/µL) and injected into unfertilized eggs.

### 2.3. Synthesis of RNAs 

We used the Gateway system (Invitrogen, Thermo Fisher, Illkirch, France) to prepare N and C-terminal fusion constructs using pSPE3::RFA::Venus, pSPE3::Venus::RFA and pSPE3::RFA::Rfp1 (a gift from P. Lemaire), plus pSPE3::Rfp1::RFA, pSPE3::RFA::mCherry and pSPE3::mCherry::RFA. All synthetic mRNAs were transcribed and capped with mMessage mMachine kit (Ambion, Thermo Fisher, Illkirch, France). Gene models and origin of all GFP-type constructs used: Ens::3GFP-NP_003971.1; Plk1::Ven-KH2012:KH.C12.238; MAP7::GFP-BC052637; EB3::3GFP-AY893969 can also be found in our methods article [20], and Kif2: phmamm.g00002556 which we characterized previously [21]. Briefly, synthetic mRNAs for the various constructs (Plk1::Ven, Ens::3GFP, MAP7::GFP, Kif2::mCherry) were microinjected into unfertilized *Phallusia* eggs which were left overnight to translate fluorescent fusion protein products [20].

The study was conducted in accordance with the Declaration of Helsinki, and since only invertebrates (ascidian and mollusc) were used no ethical declaration was required.

## 3. Results

Ascidian eggs are arrested at metaphase of meiosis I, and they extrude both PBs within a half hour of fertilization [22]. The second polar bodies were emitted at precisely the same site of the zygote surface as the first polar body causing PB1 to become tethered to the zygote surface via PB2 (*n* = 15/15, Figure 1). Careful time-lapse observations showed that a small polar corps (defined as the midbody situated between PB1 and PB2) formed on the cortical surface of a protruding outpocket that will form PB2, and also that this small polar corps linked PB1 to PB2 (Figure 1Ai and Appendix A). As a consequence of this precise spatial control, PB1 became tethered to the fertilized egg indirectly via PB2 in both ascidian and mollusc (Figure 1). To rule-out the effect of external egg coats influencing PB tethering, all experiments in the ascidian were performed using dechorionated eggs. In the complete absence of a chorion PB2 tethering to PB1 still occurred in the ascidian (Figure 1Ai,Aii). To determine the precise location of the polar corps on the cortical outpocket, we analyzed the angle between the polar corps and the apex of the outpocket (Figure 1Aii). The polar corps was centered within 9° of the cortical outpocket apex (81.4°+/− 1.4°, mean +/− sem., *n* = 15 and Figure 1Aii). We also show that PB1 is tethered to the fertilized egg surface via PB2 in the bivalve *Mytilus galloprovincialis* (Figure 1B and Appendix A). Indeed, published images of oocytes/fertilized eggs and their polar bodies from several invertebrate species show that PB1 is almost always tethered indirectly to the egg surface via PB2 (see Figure 6 model and Table 1). To be tethered to each other, PB2 must be extruded at the previous site of PB1 formation-we suggest that delocalization of the second meiotic spindle would abolish the tethering between polar bodies and instead PB1 and PB2 would each be linked to the oocyte/fertilized egg surface independently (see model, scenario 2: Figure 6). 

We sought to determine whether this small polar corps was the midbody remnant from the first meiotic division using live imaging of Polo kinase (Plk1). Plk1 is a conserved component of the spindle midzone and midbody remnant in many cell types [30,31], including mouse oocyte where it localizes to the first midbody formed during PB1 emission [32,33]. We had previously found that ascidian Plk1::Ven strongly labels the central spindle and midbodies in mitotic cells of the ascidian embryo [20]. Using live fluorescence imaging of Plk1::Ven, we followed more precisely the location of the midbody following PB1 emission (Figure 2A). The small polar corps labelled with Plk1::Ven and was present on the surface of the fertilized egg next to the site where PB1 was attached to the zygote throughout meiosis II (Figure 2A, Egg1). During PB2 emission, this small polar corps labelled with Plk1::Ven protruded from the cortical outpocket of the forming PB2 and eventually linked PB2 to PB1 (Figure 2A, arrows in Eggs 2, 3 and 4). The meiotic spindle and the midbody were labelled simultaneously by co-injecting EB3::3GFP to label microtubules together with Plk1::Rfp1 mRNA (Figure 2B). Plk1::Rfp1 again labelled the midbody that formed at the apex of PB2 outpocket (Figure 2B). Note also that Plk1::Rfp1 also labelled the chromosomes in Meta I (Figure 2B). Finally, one confocal section from a z-stack through a live fertilized egg clearly shows the position of midbody 1 between PB1 and PB2, and midbody 2 between PB2 and the zygote surface (Figure 2C and Appendix A). Plk1::Ven also labeled the midzone and the midbody in the embryo (Figure 2D).

To determine whether the midbody remnant of PB1 was located inside the fertilized egg following PB1 emission, we removed PB1 by gentle pipetting and searched for labelling of the midbody remnant in such PB1-free zygotes. We pipetted fertilized eggs during meiosis II to remove PB1, fixed and performed Phalloidin staining to observe the actin accumulated at the PB1 midbody. In fertilized eggs that had their PB1 removed, the PB1 midbody was still associated with the zygote (Figure 3A). In addition, we found recently that anti-phospho aPKC (atypical protein kinase C) strongly labels the midbody (Pruliere et al., in preparation). We thus also examined fertilized eggs that had PB1 removed with anti-phospho aPKC to label the midbody, anti-tubulin for microtubules and DAPI for DNA. The midbody remnant was again found to be located with the zygote following removal of PB1 (Figure 3B). We thus speculate that the PB1 midbody remains within the fertilized egg following PB1 emission. However, these data do not formally prove that the midbody remnant remains within the zygote following removal of PB1 since there remains the possibility that the midbody remnant is attached to the zygote surface.

Next we wished to monitor the behavior of the second meiotic spindle in order to determine its dynamics during PB2 emission. To monitor microtubules of the second meiotic spindle, we microinjected eggs with mRNA encoding Ens::3GFP mRNA (we also used MAP7::GFP or EB3::3GFP) and incubated them overnight to allow translation of the fluorescent protein [20]. We noted that the second meiotic spindle rotated during emission of PB2, with one pole anchored near the site where PB2 was emitted (Figure 4 and Appendix A). We therefore speculate that the remnant of the PB1 midbody and the subdomain of cortex polarized around the midbody attracts one pole of the second meiotic spindle, causing one pole of the spindle to enter the cortical outpocket during second meiotic spindle rotation. 

We sought to determine whether chromatin could cause polarization of the cortex in the ascidian as in the mouse [7], and also determine the consequences of chromatin-induced polarization of the cortex when the spindle failed to rotate. We observed that two cortical outpockets, one on either side of PB1, occur during failed PB2 emission (see Appendix A). During such aberrant polar body extrusion when two cortical outpockets are formed the second meiotic spindle failed to rotate (5/5 examined zygotes). Two patches of chromatin accumulated at the spindle poles (Figure 5A and Appendix A). However, and more importantly, the cortex formed protrusive outpockets above both sets of chromatids and spindle poles leading to the emission of two simultaneous PB2 outpockets (Figure 5A and Appendix A). Finally, it should be noted that chromatin alone is capable of inducing cortical polarization in mouse oocytes [7].

In this report, we show that PB2 is emitted at the site of PB1 cytokinesis. Our results show that a small polar corps forms on the surface of PB2 outpocket, and that this polar corps represents the midbody remnant formed during PB1 cytokinesis. This small polar corps linked PB1 to PB2. PB1 therefore became tethered indirectly to the fertilized egg via the polar corps on PB2. In addition, we show that during PB2 cortical outpocketing, one pole of the second meiotic spindle enters the cortical outpocket accompanied by rotation of the second meiotic spindle. Finally, we demonstrate that failure of the second meiotic spindle to rotate can lead to two simultaneous cortical outpockets rather than emission of one PB2.

## 4. Discussion

Although a Ran-GTP gradient emanating from the chromosomes of the meiotic spindle causes the overlying cortex to polarize in readiness for PB formation [7,12], we suggest that the precise location of PB2 may be dictated by the midbody remnant left behind in the fertilized egg following emission of PB1. This article is based on correlations and as such is speculative. The following discussion is therefore based on the possibility that midbody remnants influence the positioning of the second polar body. We suggest that the midbody remnant formed during PB1 formation likely remains in the fertilized egg following PB1 emission and sits at the apex of the cortical outpocket that will form PB2. This situation is similar to the finding in somatic cells where midbody remnants remain in one of the two daughter cells [34,35] rather than being externalized following cell division [36]. Once PB2 has been emitted, the midbody remnant therefore links PB1 to PB2, and thus PB1 becomes indirectly tethered to the fertilized egg surface via PB2 (see Figure 6, scenario 1). We thus use the phrase “tethered polar bodies” to reflect scenario 1 in Figure 6 whereby PB1 is tethered to the egg indirectly via PB2 (Figure 6). Please note that not all species display tethered polar bodies, and instead that both PB1 and PB2 can be linked to the egg surface directly (Figure 6, scenario 2). Due to the widespread occurrence of tethered polar bodies within the invertebrates including ascidians, we thus came to test the hypothesis that the midbody formed during emission of PB1 may direct the precise site of PB2 formation.

An analysis of published images of oocytes/fertilized eggs with two polar bodies indicates that a diverse array of species show that PB1 is tethered to PB2 instead of being linked directly to the surface of the oocyte/zygote (Figure 6, scenario 1). For example, in jellyfish PB1 is tethered to PB2 instead of being linked directly the oocyte surface [23]. Even clearer examples are offered by species belonging to the lophotrochozoa. First, in the nemertean *Micura alaskensis* PB1 is tethered to the oocyte via PB2 [24]. Likewise, in the nudibranch *Cuthona lagunae* PB1 is tethered to the oocyte via PB2 [25]. In marine bivalves, PB1 and PB2 are again tethered, for example in *Acila castrensis* (see Von Dassow, Center for Dynamics and Table 1 for link to website). We also noted that PB1 and PB2 are tethered in our time-lapse recordings of polar body emission in the mussel *Mytilus galloprovincialis* (Figure 1B and Appendix A). Among the ecdysozoa it is less clear whether PB1 is tethered to PB2. For example, in shrimp oocytes PB1 is located outside the hatching envelope far from PB2 [37]; it is also difficult to determine the precise position of polar bodies in *C.elegans* [28], although both polar bodies appear in close proximity at the anterior end of the zygote [29]. In echinoderms, starfish PB1 is tethered to PB2 and not linked directly to the oocyte surface [26], as is the case in the sea cucumber [27]. It should be noted that in many invertebrate species including annelids, nemerteans, molluscs and echinoderms that the meiotic spindles are astral and thought to possess centrioles, while the meiotic spindles in *C.elegans* and chordates (including ascidians) are not thought to possess centrioles [38]. There is therefore no correlation between centriole presence and tethered polar bodies. Moreover, even if centrioles were involved in positioning PB2 emission site, the same questions would arise: how does PB1 become tethered to PB2 instead of being directly linked to the oocyte/fertilized egg, and is the midbody remnant found between PB1 and PB2? However, exceptions to PB1 tethering are found, notably in the vertebrates (*Xenopus* and mouse), whereby both PB1 and PB2 are each linked directly to the fertilized egg surface [12,30] (and Figure 6, scenario 2). We do not know the reason for this difference; however since vertebrate oocytes arrest at Metaphase II (none of the invertebrate species listed in Table 1 arrest at Meta II) for an extended period, it is possible that this long arrest gives the chromosomes on the second meiotic spindle ample time to define the site of PB2 emission using a Ran-GTP gradient only. Furthermore, in Meta II arrested mouse oocytes some evidence indicates that the midbody formed during PB1 emission may be lost, since GFP::Plk1 labels midbody 1 during PB1 emission, but there is no remaining GFP::Plk1 labelling of the midbody remnant in Meta II arrested oocytes [33].

Some aspects of the temporal order of events during PB emission have emerged from elegant studies in *Xenopus* and mouse. Early during meiosis I in *Xenopus,* an actin cap is present before the fall in Cdk1 activity at the end of meiosis I, and it is the fall in Cdk1 that leads to the recruitment of active Cdc42 during anaphase to the actin cap causing outpocketing [14]. In *Xenopus* oocytes however, since MPF activity falls to low levels long before Cdc42 cortical recruitment is observed, it has been suggested that anaphase-specific spindle changes (occurring as a consequence of the low MPF activity) act as the precise temporal trigger for Cdc42 cortical recruitment [14]. In mouse oocytes, active Cdc42 also forms a cortical cap above one pole of the first meiotic spindle once the spindle has migrated close to the cortex [16], and a second cap of active Cdc42 is present during PB2 emission [15]. Similarly, in the mouse oocyte it had previously been shown that the cortical outpocket of PB1 was abolished by preventing the fall in Cdk1 activity, but not by preventing homologue disjunction [39,40]. Thus, in both *Xenopus* and mouse oocytes, the activation of the anaphase promoting complex (APC), which leads to the destruction of cyclin B (inactivating MPF) and securin (activating separase), is permissive for the cortical recruitment of active Cdc42 driving outpocketing during anaphase. Finally, although the fall in MPF activity is permissive for outpocketing, it is not clear how active Cdc42 is recruited to the actin cap during anaphase, although spindle pole proximity to the cortex is thought to be required [18]. Whether chromatin is also involved in driving Cdc42 recruitment to the actin cap during outpocketing (when spindle microtubules are depolymerized) is difficult to assess in mouse oocytes, because microtubule depolymerization activates the spindle assembly checkpoint (SAC) thus preventing the fall in MPF activity [41]. However, in mouse oocytes DNA beads can induce an actin cap and more importantly a cortical outpocket forms near the beads [42]. It should be noted that although microtubules were not detected around the DNA beads it is possible that some microtubules escaped detection in these immunofluorescence images [42]. Thus, in oocytes it is not entirely known whether chromatin and/or the spindle pole triggers Cdc42 accumulation at the actin cap triggering outpocketing.

Based on the results presented here, we speculate that the midbody remnant is located at the site where the actin cap forms during meiosis II, and that this cortical site prefigures the location of cortical outpocketing during PB2 emission. For example, our evidence indicates that actin is located around the midbody early during meiosis II (perhaps in part due to a Ran-GTP gradient coming from the adjacent chromosomes activating Cdc42) at the site where cortical outpocketing will later occur during Anaphase II. We also speculate that for efficient PB2 emission one pole of the second meiotic spindle may be attracted to the midbody remnant, and that during cortical outpocketing this may be involved in spindle rotation as the free spindle pole is forced to rotate into the egg interior while the other spindle pole rotates into the PB outpocket. In addition, we speculate that the rotation of the spindle which moves one spindle pole and its associated chromatids away from the cortex may be involved in creating only one cortical outpocket rather than two. Indeed, when the spindle fails to rotate two simultaneous cortical outpockets are formed, one on either side of PB1 (Figure 5). However, we still do not know how one pole of the spindle is captured by the polarized subdomain of cortex and midbody remnant. Interestingly, in *C.elegans* 2-cell stage embryos, the midbody remnant that remains in the P1 daughter cell following cytokinesis influences astral microtubules during nucleus-centrosomal complex rotation thus biasing the orientation of the mitotic spindle in the P1 cell [35]. Also, it has recently been demonstrated in preimplantation mouse embryos that the cytokinetic bridge connecting two sister blastomeres acts as a scaffold leading to microtubule stabilization and outgrowth during interphase [43]. So microtubules could potentially grow from either the spindle pole or even from the midbody remnant to influence second meiotic spindle position (however, we do not discount the involvement of an actin-based mechanism). It should be noted that the identity of these microtubules is not known, and indeed in the ascidian egg the meiotic spindle is not thought to possess centrioles or astral microtubules. However, the absence of centrioles does not preclude the presence of short astral microtubules. For example, in *Xenopus* and *C.elegans* oocytes that also lack centrioles, the meiotic spindle poles displays short astral microtubules [44,45]. Unfortunately, due to the density of microtubules in the meiotic spindle it is difficult to detect astral microtubules in the ascidian. Nonetheless, since tethered polar bodies are a widespread occurrence throughout the invertebrates, we speculate that the site of the previous PB1 cytokinesis may direct the precise positioning of PB2 formation (Table 1). We also wonder whether the occurrence of tethered polar bodies is linked to rapid progression though meiosis II in oocytes that do not arrest at Meta II (the vast majority of invertebrates: exceptions are chaetognaths and amphioxus). Meiosis II lasts only 15 min. in the ascidian. In species that do not display tethered polar bodies such as the vertebrates, oocytes can remain arrested at Meta II before fertilization for several hours. Finally, this proposition is somewhat similar to the well-studied case of budding yeast, in which each new daughter cell emerges adjacent to the previous cytokinetic ring (« bud scar ») [46].

## 5. Conclusions

Overall, in the ascidian we show that PB1 is tethered to the fertilized egg via PB2 and that a small polar corps containing the midbody remnant is situated between PB1 and PB2. We speculate that the midbody remnant formed during PB1 cytokinesis may remain inside the zygote and possibly attract one pole of the spindle thus aiding in the rotation of the second meiotic spindle during PB2 emission. We also show here that when the second meiotic spindle fails to rotate in poor quality zygotes the emission of PB2 is not normal. For example, this is associated with two outpockets forming (one on either side of PB1) rather than one outpocket adjacent to PB1. Interestingly, in some species of Clam (*Corbicula leana*) failure to rotate the meiotic spindle is a natural event. For example, in *Corbicula leana* two first polar bodies form simultaneously leading to a complete loss of the egg chromosomes thus causing androgenesis of the fertilized egg [47]. Finally, we point out the obvious limitation of the current work which is mostly descriptive and correlative due to our inability to either destroy or remove the midbody remnant without also destroying or removing either the actin cap or the egg chromosomes respectively.

## Figures and Tables

**Figure 1 genes-11-01394-f001:**
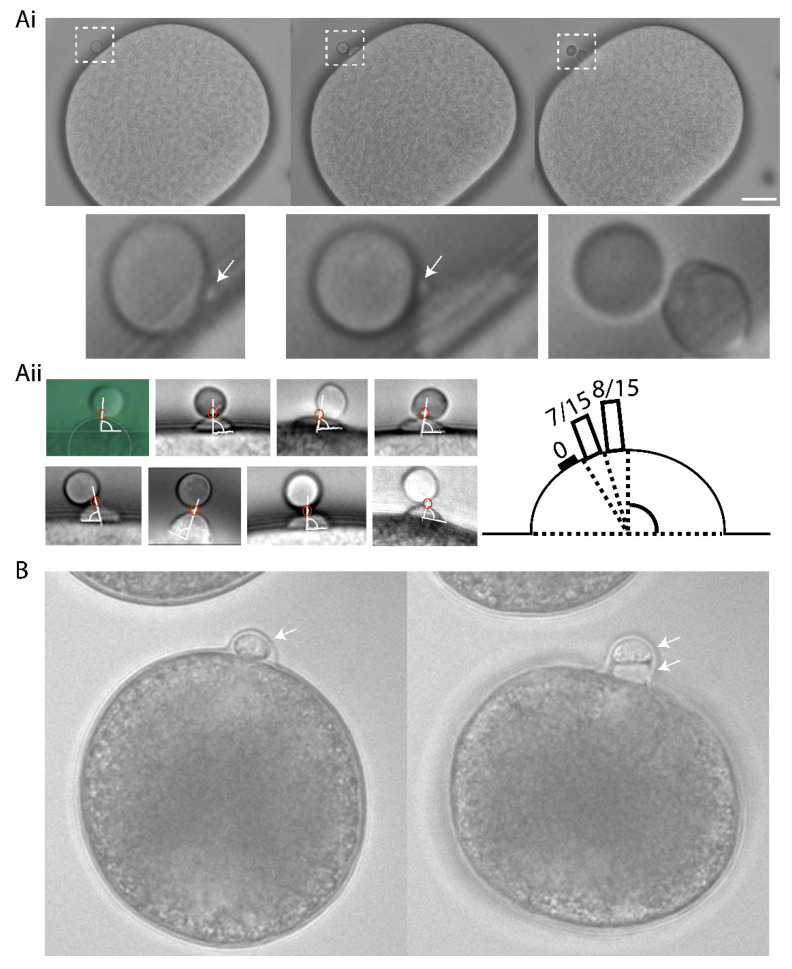
A small protrusion forms on the surface of the PB2 outpocket in *Phallusia*. (**Ai**) Bright field images from a time-lapse experiment of fertilized *Phallusia mammillata* eggs. During polar body 2 (PB2) emission, a small polar corps can be observed on the surface of PB2 outpocket (boxed region). Enlarged views of the boxed regions at the bottom show in greater detail the small protrusion which is present before outpocketing begins (bottom left, arrow) and remains present during outpocketing (bottom middle, arrow). *n* > 50 eggs. Scale bar = 20 µm. See Appendix A. (**Aii**) Analysis of angle between polar corps (red circle) and outpocket center. Diagram illustrating a summary of the polar corps position at 61–70°, 71–80° and 81–90° (number of eggs in brackets). Mean polar corps position was 81.4°+/−1.4° +/−sem, *n* = 15. (**B**) Bright field images from a time-lapse experiment showing PB2 emission under PB1 in *Mytilus galloprovincialis.* PB1 is indicated by the arrow in the first image, and both PB1 and PB2 by the two arrows in the second image. *n* = 50/50. See Appendix A.

**Figure 2 genes-11-01394-f002:**
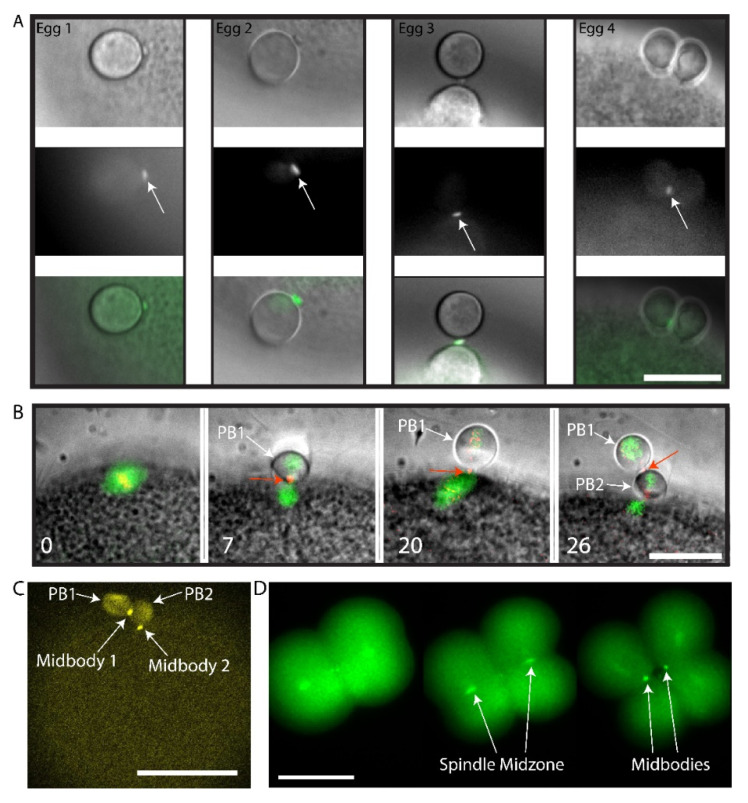
Plk1::Venus labels the midbody that forms between PB1 and the egg. (**A**) Four different examples of the small polar corps labelled with Plk1::Ven. Upper row of images shows bright field images of PB2 emission site adjacent to PB1. Middle row of images shows that Plk1::Ven labelled the midbody that formed between PB1 and the egg (see arrows). Bottom row is the overlay. Plk1::Ven localization to the midbody during the process of PB2 emission (see arrows). *n* = 12/12. Scale bar = 20 µm. (**B**) Epifluorescence images of meiotic spindle labelled with EB3::3GFP and the midbody with Plk1::Rfp1. Please note that Plk1::Rfp1 labels the chromosomes (red) on the Meta I spindle (first image), then the midbody (second image) and also that the midbody is found at the apex of the PB2 outpocket (third image). PB1 is tethered to PB2 (fourth image). First midbody is indicated with red arrows and PBs are indicated by the white arrows. Time in minutes is indicated. Scale bar = 20 µm. *n* = 5/5 from time-lapse experiments. (**C**) Fertilized egg after PB1 and PB2 emission. One confocal plane from a z-stack showing the localization of Plk1::Ven to midbody 1 and midbody 2 (arrows). PB1 and PB2 are also indicated by arrows. *n* = 12/12. Scale bar = 50µm. See Appendix A. (**D**) Two to four cell-stage. Epifluorescence images of midbody formation. Plk1::Ven labels the central spindle (arrows) then both midbodies (arrows) at the end of cytokinesis. *n* = 22. Scale bar = 50 µm.

**Figure 3 genes-11-01394-f003:**
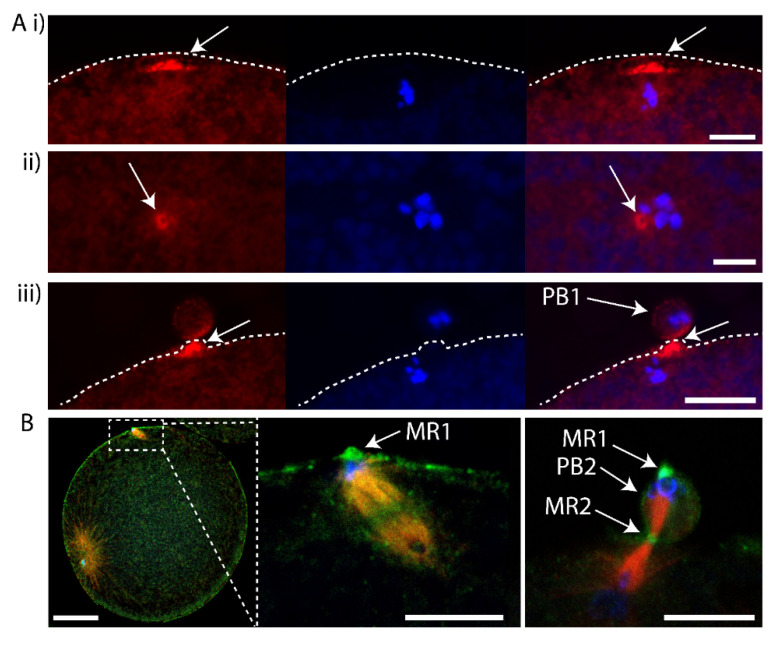
First midbody remains associated with the fertilized egg following PB1 emission. (**A**) Fertilized eggs were fixed during meiosis II and stained using Phalloidin::TRITC to label actin and Hoechst to label chromosomes. (i) Confocal images showing actin cap (arrows) during meiosis II in an egg following PB1 removal. (ii) Confocal images showing actin labelling of the midbody (arrows) during meiosis II in an egg following PB1 removal (arrows). (iii) Control egg that displayed PB1 attached to the egg surface. Please note that the midbody is strongly labelled with Phalloidin (arrows). Dotted line indicates surface of the egg. Scale bars = 10 µm. *n* > 50 for (i) and (ii) where Phalloidin labelled the actin cap (i) and midbody (ii). *n* > 50 for Phalloidin labelling of outpocket (iii). (**B**) Fertilized eggs were pipetted during meiosis II to remove PB1, fixed and labelled with anti-phospho aPKC (green), anti-tubulin (red) and DAPI (blue). Left image: overlay showing the rotated second meiotic spindle. The sperm aster is also visible, far left. Scale bar = 30µm. Middle image: inset of boxed region showing that one pole of second meiotic spindle is aligned with PB1 midbody remnant (MR1 arrow). Please note that PB1 was removed by pipetting. Scale bar = 10µm. Right image: Another fertilized egg which had already emitted PB2 showing location of first midbody remnant (MR1, arrows), second midbody remnant (MR2, arrow) and PB2 (arrow). Please note that PB1 was removed by pipetting. *n* > 50 zygotes where aPKC labelled the midbody remnant. Scale bar = 10µm. Also see Appendix A of LifeAct labelling.

**Figure 4 genes-11-01394-f004:**
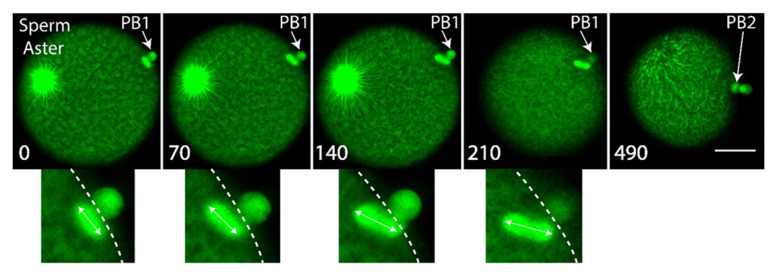
Second meiotic spindle rotates during PB2 emission. Eggs were previously microinjected with mRNA encoding Ens::3GFP to label the microtubules (green). Confocal images extracted from a time-lapse experiment showing the rotation of the second meiotic spindle. Upper row of images showing that the second meiotic spindle lies under PB1 parallel to the cortical surface, then begins to rotate (image 2) and continues to rotate (images 3 and 4) as PB2 is emitted (see last image of Figure and of Appendix A). The meiotic spindle rotates 50°+/− 3°, *n* = 13, mean +/− sem. Insets at the bottom show more clearly the rotation of the second meiotic spindle (double headed arrow shows spindle orientation). Large sperm aster is also visible. n = 13. Scale bar = 40 µm. Time between images is indicated in seconds on each image. See Appendix A.

**Figure 5 genes-11-01394-f005:**
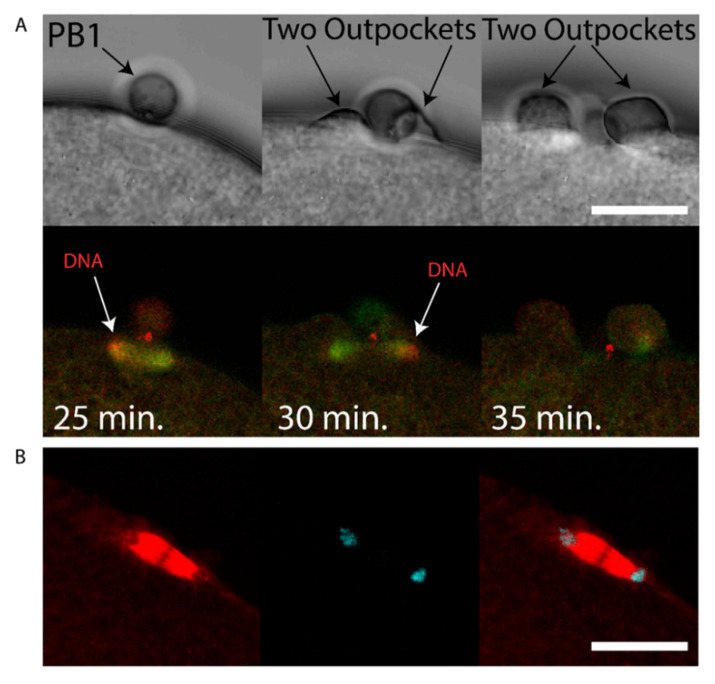
Failed rotation of second meiotic spindle giving two PB2 outpockets. (**A**) Unfertilized eggs were injected with mRNAs encoding Ens::3GFP (microtubules green) and Kif2::mCherry (chromosomes red) mRNA. Two outpockets are shown in the second and third bright field images respectively (arrows). Please note that the second meiotic spindle fails to rotate and two outpockets form above both sets of chromosomes following Ana II (see 24–31 min. on Appendix A). Note also that Kif2::mCherry also labels the midbody (prominent red staining near PB1). Scale bar = 20 µm. *n* = 5/5 examples of parallel spindles. See Appendix A for the full data-set. Another example of two simultaneous PB2 outpockets is shown in Appendix A. (**B**) Fertilized eggs were fixed and labelled with anti-Tubulin (red) and stained with DAPI (blue). Confocal images of a spindle positioned parallel to the egg cortex in anaphase. Scale bar = 20 µm, *n* = 5 examples of parallel MII spindles.

**Figure 6 genes-11-01394-f006:**
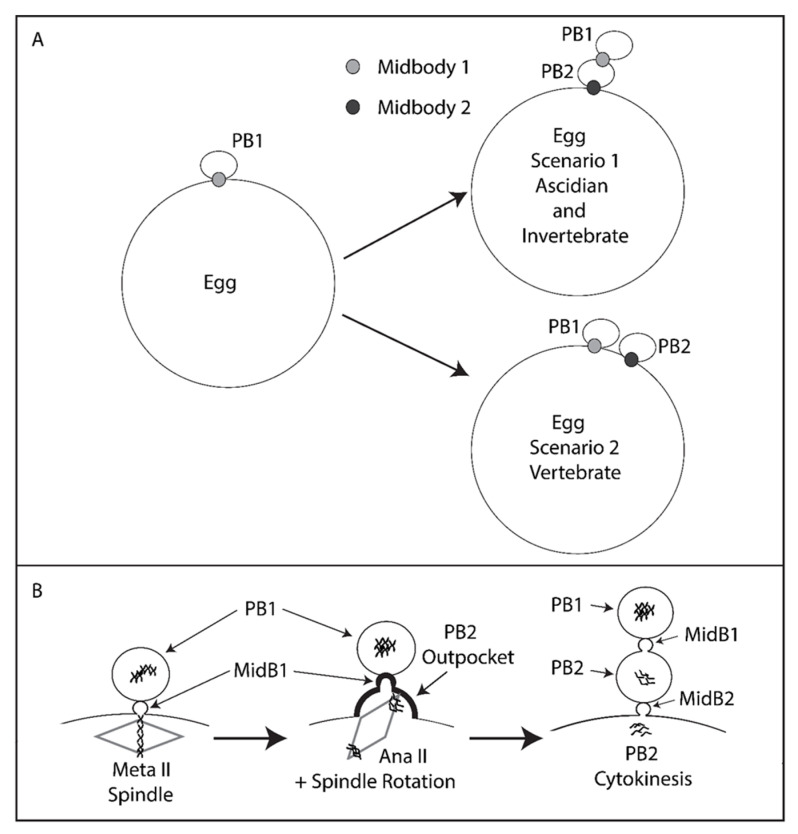
Model. Tethered polar bodies. (**A**) Scenario 1: tethered polar bodies. Following emission of both polar bodies, the first midbody attaches PB1 to PB2 while the second midbody links PB2 to the egg. PB1 is thus tethered to the egg indirectly via PB2. Scenario 1 represents ascidians and possibly many invertebrates (see Table 1 for details). Scenario 2: the first midbody links PB1 to the egg and the second midbody also links PB2 to the egg. Scenario 2 represents mouse and *Xenopus*. Midbody 1 is depicted as light grey, midbody 2 as dark grey. However, it should be stressed that midbody position is only known for certainty in the ascidian (this study). (**B**) PB2 emission dynamics. Midbody 1 is located at the apex of PB2 cortical outpocket. Spindle rotation into PB2 cortical outpocket is displayed. PB2 is emitted attached to PB1 via midbody 1 (MidB1) and PB1 is thus tethered indirectly to the egg via PB2.

**Table 1 genes-11-01394-t001:** Tethered polar bodies. Several species display tethered first and second polar bodies as in the ascidian and depicted in Scenario 1. This is not an exhaustive list since we noted several other examples that are not detailed here. Notable exceptions are the vertebrates that do not show tethered polar bodies. * Center for Cell Dynamics website: http://rusty.fhl.washington.edu/celldynamics/gallery/index.html.

Species	PBs Tethered	Publication
Jellyfish *Clytia hemispherica*	Yes	[23]
Nemertean worms *Micura alaskensis*	Yes	[24]
Sea slug *Cuthona lagunae*	Yes	[25]
Clam * *Acila castrensis**	Yes	Von Dassow * Center for Cell Dynamics website *
Mussel *Mytilus galloprovincialis*	Yes	This study
Starfish *Asterina pectinifera*	Yes	[26]
Sea cucumber *Holothuria moebi*	Yes	[27]
Ascidian *Phallusia mammillata*	Yes	This study
Nematode *Caenorhabditis elegans*	Unclear	[28,29]
*Xenopus*	No	[30]
Mouse	No	[15]

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
