# Peer review of "Role of PB1 Midbody Remnant Creating Tethered Polar Bodies during Meiosis II"

_genes, 2020, doi:10.3390/genes11121394_

Round 1

Reviewer 1 Report

Polar body formation is an essential process in many animals because it is required to reduce chromosome number in oocytes to prevent polyploidy.  This manuscript thus addresses an important general area but the specific contribution of this manuscript is less clear.  The manuscript contains mostly descriptive data suggesting that one pole of the ascidian meiosis II spindle is attached to the midbody remnant from meiosis I and that after rotation of the meiosis II spindle, this results in tethering of polar body I to polar body II.  Although interesting, the manuscript is not publishable in it current form for the following major reasons.  First, even with an extremely lengthy discussion, there is no suggestion of a testable hypothesis why tethering of polar bodies might be significant.  This will frustrate even the most avid afficionado of polar body extrusion. Second, the writing needs to be clarified so that the reader can understand what is happening.  The use of the undefined word “corps” in the abstract and first paragraph of results is extremely confusing.  Third, all of the data needs to be presented in a quantitative manner with n given.  Fourth, the nocodazole experiment is not publishable (see details).

Detailed comments

The word “corps” is not defined in the abstract.  The abstract is extremely long but does not explain the significance of tethered polar bodies.  No functional experiments removing the midbody are described in the abstract.

Line 62 states that ran-GTP-mediated outpocketing is a difference between polar body formation and mitotic cytokinesis but cites no reference showing normal cytokinesis in the complete absence of GTP-ran.  Chromosome-induced out-pocketing would occur symmetrically over each spindle pole in mitosis.

Line 83, there is a huge logical jump from the unknown mechanism targeting cdc42 to midbody 1 determining the position of polar body 2.  This reviewer did not grasp this connection until the end of the paper.   There is no suggestion in the introduction why a reader should care about polar bodies being tethered together.  This will make it difficult for the most enthusiastic reader to continue reading.

The beginning of the results uses the word “corps” without defining it.  The first half of the legend to figure 1 uses the word “protrusion” then switches to “corps”.  Replacing the word corps with protrusion would make the manuscript clearer.

Line 133: “PB1 became tethered to the egg indirectly via PB2”, There is no data shown to support this statement.  There is no data supporting the same statement on line 137.  In Fig. 1B, there is clearly a viteline envelope that would prevent an untethered polar body from floating away.  Definitively demonstrating the absence of any viteline envelope or fertilization envelope in Fig. 1A would help bolster the argument that the position of a polar body is evidence that it is tethered at that position.  Results presented in Fig. 3 indicate that gentle pipetting removes PB1, suggesting it was not tethered to PB2.

Line 129: “These two PBs are always emitted” Please replace with: “These two PBs were emitted in X/Y cases” OR cite a reference that defines “always” with numbers.

Data in figure 1B needs n as well.

Line 141: “delocalization by as little as about half the length of the second meiotic spindle would abolish the tethering between polar bodies”, This appears to be speculation rather than a result. 

If Fig 2B is a time-lapse sequence of the same zygote, please put add time stamps and give n for the number of time-lapse sequences that gave this result.  If fig. 2C is a maximum intensity projection, please state this in the legend and give n for the number of z-stacks that yielded the same result.  Also please state how many images were projected and the space between them in um.

Please give n for the results in Fig. 3A, B and C.

Abscission can occur on one side, both sides or neither side of a midbody.  The results in Fig. 3B (if quantified) suggest that abscission does not occur on the egg side of midbody 1, thus leaving a structure that could be attached to one pole of the meiosis II spindle.  This could be more clearly explained.  The cartoon in Fig. 7 implies that abscission did occur on the side of midbody 1 distal to the egg but this is not clear from the data shown.  Did “gentle pipetting” tear a cytoplasmic bridge or did it simply remove a viteline or fertilization envelope?

PB2 formation is not shown in Fig. 4 but it is stated as occurring in the text.  Are there corresponding brightfield images to support the statements in the results text? N=12 is written in the legend to Fig. 4 but the text should clearly state what happened in 12/12 cases.  Fig. 4 needs time stamps if it is indeed a time-lapse sequence.

“on occasion, the second meiotic spindle failed to rotate”  “on occasion” must be replaced with a numerical frequency (X/Y).  Also, the authors have suddenly switched from “tilted” to “rotated”.  “Rotated” is more commonly used in the spindle positioning field, however, if the authors wish to use “tilted”, they need to stick to the same nomenclature.

Fig. 6 is an attempt to show that chromatin can induce outpocketing of the cortex in the absence of microtubules.  Interpretation of this experiment requires definitive documentation that microtubules were completely depolymerized by the nocodazole.  There is still obvious MAP7::GFP fluorescence around the chromatin.  The legend refers to this as “a depolymerized mass”.  This result is completely unconvincing without better evidence of complete microtubule depolymerization.  The panels in this figure need to be labelled and a negative control with solvent and no nocodazole needs to be included, especially since this experiment is live imaging with Hoechst.

Table 1: Polar bodies are not tethered to each other in C. elegans.  They are separated by egg shell layers that are secreted sequentially during meiosis and are separated by long distances by the first cleavage.

Author Response

We thank both reviewers for their helpful comments which have greatly improved the manuscript. All changes in the main text of the article are highlighted in yellow. Please note the extensive re-writing as suggested by the reviewers to “tone down” the claims to indicate that this article is speculative. Specific replies to specific points are detailed below.

Review 1

Comments and Suggestions for Authors

Polar body formation is an essential process in many animals because it is required to reduce chromosome number in oocytes to prevent polyploidy.  This manuscript thus addresses an important general area but the specific contribution of this manuscript is less clear.  The manuscript contains mostly descriptive data suggesting that one pole of the ascidian meiosis II spindle is attached to the midbody remnant from meiosis I and that after rotation of the meiosis II spindle, this results in tethering of polar body I to polar body II.  Although interesting, the manuscript is not publishable in it current form for the following major reasons.  First, even with an extremely lengthy discussion, there is no suggestion of a testable hypothesis why tethering of polar bodies might be significant. 

We have added our speculation at the end of the introduction:

Here in the ascidian we suggest that during meiosis II the midbody formed during PB1 emission predicts the precise site of PB2 outpocketing and we speculate may be involved in specifying the precise location of PB2.

This will frustrate even the most avid afficionado of polar body extrusion. Second, the writing needs to be clarified so that the reader can understand what is happening.  The use of the undefined word “corps” in the abstract and first paragraph of results is extremely confusing. 

The polar corps has now been defined in the abstract:

During outpocketing of PB2 in ascidians, we discovered that a small structure around 1µm in diameter protruded from the cortical outpocket that will form the future PB2, which we define as the “polar corps”.

Third, all of the data needs to be presented in a quantitative manner with n given. 

Precision on n numbers has now been added throughout (see Figure legends and answers to precise questions below.

 Fourth, the nocodazole experiment is not publishable (see details).

This figure has been removed as suggested.

Detailed comments

The word “corps” is not defined in the abstract.  The abstract is extremely long but does not explain the significance of tethered polar bodies.  No functional experiments removing the midbody are described in the abstract.

The abstract has been amended to define the “corps”:

During outpocketing of PB2 in ascidians, we discovered that a small structure around 1µm in diameter protruded from the center of the cortical outpocket that will form the future PB2, which we define as the “polar corps”.

Line 62 states that ran-GTP-mediated outpocketing is a difference between polar body formation and mitotic cytokinesis but cites no reference showing normal cytokinesis in the complete absence of GTP-ran.  Chromosome-induced out-pocketing would occur symmetrically over each spindle pole in mitosis.

Text has been changed to: “One essential finding therefore is that a chromatin-centered Ran-GTP gradient causes a subdomain of the cortex adjacent to the meiotic chromosomes to become polarized driving PB outpocketing (Dehapiot et al., 2013; Deng et al., 2007).”

Line 83, there is a huge logical jump from the unknown mechanism targeting cdc42 to midbody 1 determining the position of polar body 2.  This reviewer did not grasp this connection until the end of the paper.   There is no suggestion in the introduction why a reader should care about polar bodies being tethered together.  This will make it difficult for the most enthusiastic reader to continue reading.

Text has been changed to : Thus, cortical outpocketing during anaphase I and II is thought to be induced by the cortical accumulation of active Cdc42 in Xenopus and mouse oocytes. However, it is not entirely clear what triggers the accumulation of active Cdc42 at the cortex driving outpocketing (Leblanc et al., 2011), and indeed how one outpocket rather than two are formed during meiosis II.  Here in the ascidian we have discovered that during meiosis II the midbody formed at PB1 emission predicts the precise site of PB2 outpocketing.

The beginning of the results uses the word “corps” without defining it.  The first half of the legend to figure 1 uses the word “protrusion” then switches to “corps”.  Replacing the word corps with protrusion would make the manuscript clearer.

Text has been changed to :

We have clarified this confusion in the text/introduction, paragraph 2: The protrusive outpocket therefore defines the bulge which will form the future polar body.

Also, at the end of the introduction we have added:

Here in the ascidian we found that during meiosis II the midbody formed during PB1 emission becomes visible as a small “polar corps” sitting on the protrusive outpocket and that this polar corps predicts the precise site of PB2 outpocketing; we thus speculate that the polar corps may be involved in specifying the precise location of the PB2 protrusive outpocket.

In the Results section, paragraph 1: Careful time-lapse observations showed that a small polar corps (defined as the midbody situated between PB1 and PB2) formed on the cortical surface of a protruding outpocket that will form PB2, and also that this small polar corps linked PB1 to PB2 (Figure 1Ai and Supp Movie 1).

Line 133: “PB1 became tethered to the egg indirectly via PB2”, There is no data shown to support this statement.  There is no data supporting the same statement on line 137.  In Fig. 1B, there is clearly a viteline envelope that would prevent an untethered polar body from floating away.  Definitively demonstrating the absence of any viteline envelope or fertilization envelope in Fig. 1A would help bolster the argument that the position of a polar body is evidence that it is tethered at that position.  Results presented in Fig. 3 indicate that gentle pipetting removes PB1, suggesting it was not tethered to PB2.

The reviewer is correct to point-out the presence of a viteline coat in mollusc. However, there is no vitelline coat in the ascidian. The text has been clarified.  As a consequence of this precise spatial control, PB1became tethered to the egg indirectly via PB2 in both ascidian and mollusc (Figure 1). To rule-out the effect of external egg coats influencing PB tethering all experiments in the ascidian were performed using dechorionated oocytes. In the complete absence of a chorion PB2 tethering to PB1 still occurred in the ascidian (Figure 1Ai, Aii).

Line 129: “These two PBs are always emitted” Please replace with: “These two PBs were emitted in X/Y cases” OR cite a reference that defines “always” with numbers.

We agree with the reviewer that our sentence was vague and confusing. It has been changed to:

The second polar bodies were emitted at precisely the same site of the egg surface as the first polar body causing PB1 to become tethered to the zygote surface via PB2  (n=15/15, Figure 1).

Data in figure 1B needs n as well.

Figure 1B legend has been changed to : N=50/50.

Line 141: “delocalization by as little as about half the length of the second meiotic spindle would abolish the tethering between polar bodies”, This appears to be speculation rather than a result. 

We agree with the reviewer, the text has been changed to reflect the speculation: we speculate that delocalization by as little as about half the length of the second meiotic spindle would abolish the tethering between polar bodies and instead PB1 and PB2 would each be linked to the egg surface independently (see model, scenario 2: Figure 7).

If Fig 2B is a time-lapse sequence of the same zygote, please put add time stamps and give n for the number of time-lapse sequences that gave this result. 

The Figure and legend have been amended:

Time in minutes is indicated. Scale bar = 20µm. n=5 time-lapse experiments.

If fig. 2C is a maximum intensity projection, please state this in the legend and give n for the number of z-stacks that yielded the same result.  Also please state how many images were projected and the space between them in um.

This is not a maximum z-projection but rather one z-section – this has now been clarified in the figure legend. The full dataset can be visualized in Supp. Movie 3.

The legend has been amended to make this point clear:

One confocal plane from a z-stack is displayed.

Please give n for the results in Fig. 3A, B and C.

N numbers are now stated in the Figure legends for figure 3A, B and C

Abscission can occur on one side, both sides or neither side of a midbody.  The results in Fig. 3B (if quantified) suggest that abscission does not occur on the egg side of midbody 1, thus leaving a structure that could be attached to one pole of the meiosis II spindle.  This could be more clearly explained.  The cartoon in Fig. 7 implies that abscission did occur on the side of midbody 1 distal to the egg but this is not clear from the data shown.  Did “gentle pipetting” tear a cytoplasmic bridge or did it simply remove a viteline or fertilization envelope?

Ascidian do not have a vitelline envelope. The text has been clarified to make this point clear:

As a consequence of this precise spatial control, PB1became tethered to the egg indirectly via PB2 in both ascidian and mollusc (Figure 1). To rule-out the effect of external egg coats influencing PB tethering all experiments in the ascidian were performed using dechorionated oocytes. In the complete absence of a chorion PB2 tethering to PB1 still occurred in the ascidian (Figure 1Ai, Aii).

PB2 formation is not shown in Fig. 4 but it is stated as occurring in the text.  Are there corresponding brightfield images to support the statements in the results text? N=12 is written in the legend to Fig. 4 but the text should clearly state what happened in 12/12 cases.  Fig. 4 needs time stamps if it is indeed a time-lapse sequence.

Time stamps plus the corresponding PB2 image are now indicated in revised figure.

Figure 4. Legend: Two additional pieces of information have been provided in the legend of Figure 4. Time stamps plus angle of rotation.

Time between images is indicated in seconds on each image.

The meiotic spindle rotates from a parallel orientation by 50°+/- 3°, n=13, mean +/- sem

“on occasion, the second meiotic spindle failed to rotate”  “on occasion” must be replaced with a numerical frequency (X/Y).  Also, the authors have suddenly switched from “tilted” to “rotated”.  “Rotated” is more commonly used in the spindle positioning field, however, if the authors wish to use “tilted”, they need to stick to the same nomenclature.

On occasion has been replaced with:

During such aberrant polar body extrusion when two cortical outpockets are formed the second meiotic spindle failed to rotate (5/5 examined zygotes).

All mention of tilted has been replaced by rotated throughout.

Fig. 6 is an attempt to show that chromatin can induce outpocketing of the cortex in the absence of microtubules.  Interpretation of this experiment requires definitive documentation that microtubules were completely depolymerized by the nocodazole.  There is still obvious MAP7::GFP fluorescence around the chromatin.  The legend refers to this as “a depolymerized mass”.  This result is completely unconvincing without better evidence of complete microtubule depolymerization.  The panels in this figure need to be labelled and a negative control with solvent and no nocodazole needs to be included, especially since this experiment is live imaging with Hoechst.

This figure has been removed, and all reference to it in the text equally removed:

Page 9, Line 7 Removed; This is similar to our findings using ascidian eggs: we found that following microtubule depolymerization, which disrupts the spindle but does not entirely remove all tubulin staining from around the chromosomes (Figure 6), the female chromosomes can still induce a cortical outpocket that resorbs. It should be noted that ascidian eggs do not possess a SAC (Chenevert et al., 2020), so the MPF activity declines despite a depolymerized spindle (the eggs form a pronucleus at the correct time) thus allowing us to detect the cortical outpocket when MPF activity falls to low levels (Dumollard et al., 2011).

Table 1: Polar bodies are not tethered to each other in C. elegans.  They are separated by egg shell layers that are secreted sequentially during meiosis and are separated by long distances by the first cleavage.

We were also uncertain of this question in C. elegans and used the word “possibly” in Table 1. Due to the reviewers and our own uncertainty we have changed the word possibly to unclear in Table 1, and also now added the Schumacher reference in addition to the Dorn et al., 2010 reference so that readers can be directed to those articles. Although polar bodies may move during cytokinesis we have focused here on their site of emission., and thus cited the Schumacher et al., (1998) article which shows in Figure 3, panels (Q and R) that:  Polar bodies are extruded after each meiotic division at the anterior of the embryo. From that figure PB1 and PB2 are juxtaposed.

Reviewer 2 Report

In this article, McDougall and colleagues explore the tethering of polar bodies during meiosis in the ascidian Phallusia mammillata. They show that a small structure, which they identify as the midbody remnant, tethers the polar body 1 and polar body 2 during the 2 successive meiotic divisions. They also notice the same phenomenon in a mollusc oocyte and observe a similar tethering in many published images of oocytes. This is a very interesting observation. They go on to investigate a few features of polar body extrusion, such as the presence of an actin cap and the fact that there can be 2 outpocketting of the cortex if the spindle fails to rotate. Overall, this is an interesting study that has one major finding, the midbody tethering of the polar body. A major problem with the study (detailed below) is that most of the processes described (cortex polarisation, PB extrusion location, spindle rotation) correlate with the location of the midbody but there is no evidence that the midbody is instructive. Overall, I still recommend the manuscript for publication, as long as the authors make it very clear in the manuscript (and not only at the end) that this is a series of interesting correlated observations but that causation is totally speculative at this point. Similarly, I would like to see more quantifications of what is described, as it is hard to form conclusions based on 1 movie, even if the authors state n=XX.

Major comments:

  • The paper is crucially lacking quantifications. How often are each of the phenomenon described observed? If the authors want to claim correlations between the position of different structures (actin cap/midbody/PB1 or PB2 extrusion), they should quantify and correlate the relative positions. Another example is figure 4. The authors make a claim that the spindle moves along the cortex and tilts, but only show one image, they should quantify that (for example to support their claim, they could show that the spindle shifts over a distance that is half the length of the spindle, thus allowing it to allow the extrusion of the polar body 2 just under polar body 1). In figure 5 and 6, the authors show that chromatin proximity is sufficient to induce outpocketing, similarly to mouse oocytes, but only show 2 images (one in the figure, one in supplementary). They should quantify it.

  • The authors make the point that the midbody is responsible for the position of the outpocketing of the second polar body. (For example with sentences such as “Here in the ascidian we have discovered that during meiosis II the midbody formed at PB1 emission participates in directing the precise site of PB2 outpocketing” in the introduction or “we propose here in ascidians that the precise location of PB2 is dictated by the midbody remnant left behind in the egg following emission of PB1” or “we thus came to test the hypothesis that the midbody  formed during emission of PB1 may direct the precise site of PB2 formation” in the discussion.). As they state at the end of the manuscript, this is purely correlative. It could be that the midbody is retained at the site of maximum curvature (which could be quantified), that the midbody is retained by the actin cap which in turn controls where the outpocketing is… They should tone down their claim. Alternatively, they could try to demonstrate that the midbody is instructive to the position of the outpocketing. Is it not possible to do midbody preps such as Addi et al., 2020 and add midbodies in suspension to the media? This reviewer understands that it might be very difficult to get the midbodies out of the oocytes, but they could use separate oocytes, or embryos, or even adult tissues? Similarly, the statement in the discussion “our evidence indicates that actin is recruited around the midbody early during meiosis II” seems like an overclaim. It just seems like the midbody sits near the actin cap; the relationship between the two is not clear, and causality could only be demonstrated by moving the midbody (perhaps using a microneedle, or by treating the oocytes with a midbody solution).

The authors also suggest that the midbody could be responsible for spindle rotation and envision a variety of possible scenario. Again, they should tone down their claims as it is not clear whether the midbody has any influence on spindle positioning. Thus, the discussion is widely hypothetical, and it should be clear that it is.

  • The evidence that the midbody is within the oocyte is not convincing. As it stands, the staining show that 1) there is cortical actin near the midbody 2) the midbody is closely associated with both PB and oocyte. Thus, it is not clear whether the midbody is release through abscission on both sides then recaptured by the oocyte, or only severed on the polar body side, or, more likely, not severed at all, explaining the tethering. As it is not clear exactly what happens here in terms of midbody release, the authors should either test what they are claiming (by providing evidence that the midbody is within the oocyte, for example by providing membrane staining, electron microscopy images, or live imaging of abscission on the PB side for example), or tone down their claim. Similarly, in the discussion, the authors state: “This situation is similar to the finding in somatic cells where midbody remnants remain in one of the two daughter cells (Chen et al., 2013; Guizetti and Gerlich, 2010) rather than being externalized following cell division (Crowell et al., 2014).” These papers do not really say that. Guizetti and Gerlich does not mention the fate of the midbody. Crowell describes what happens after the midbody gets released then recaptured by a cell.

  • The figures should be presented to be easier to read: for example, the stainings should be indicated on the panel and not only in the legends.

Minor comments:

the abstract should state that this study is in ascidians, it is not clear at first glance.

It is not clear what figure 3A-C brings to the study.

species name should be in italic.

The methods should be readable without having to go back to a previous paper (avoid “as described previously”

Author Response

We thank both reviewers for their helpful comments which have greatly improved the manuscript. All changes in the main text of the article are highlighted in yellow. Please note the extensive re-writing as suggested by the reviewers to “tone down” the claims to indicate that this article is speculative. Specific replies to specific points are detailed below.

Review 2

Comments and Suggestions for Authors

In this article, McDougall and colleagues explore the tethering of polar bodies during meiosis in the ascidian Phallusia mammillata. They show that a small structure, which they identify as the midbody remnant, tethers the polar body 1 and polar body 2 during the 2 successive meiotic divisions. They also notice the same phenomenon in a mollusc oocyte and observe a similar tethering in many published images of oocytes. This is a very interesting observation. They go on to investigate a few features of polar body extrusion, such as the presence of an actin cap and the fact that there can be 2 outpocketting of the cortex if the spindle fails to rotate. Overall, this is an interesting study that has one major finding, the midbody tethering of the polar body. A major problem with the study (detailed below) is that most of the processes described (cortex polarisation, PB extrusion location, spindle rotation) correlate with the location of the midbody but there is no evidence that the midbody is instructive. Overall, I still recommend the manuscript for publication, as long as the authors make it very clear in the manuscript (and not only at the end) that this is a series of interesting correlated observations but that causation is totally speculative at this point. Similarly, I would like to see more quantifications of what is described, as it is hard to form conclusions based on 1 movie, even if the authors state n=XX.

Major comments:

  • The paper is crucially lacking quantifications. How often are each of the phenomenon described observed? If the authors want to claim correlations between the position of different structures (actin cap/midbody/PB1 or PB2 extrusion), they should quantify and correlate the relative positions.

      Figure 3A has been amended as follows: (Part A has been removed as required).

      n>50 for i) and ii) where Phalloidin labelled the actin cap (i) midbody (ii).

      n>50  or Phalloidin labelling of outpocket (iii).

      Figure 3B has been amended as follows:

      n>50 zygotes where aPKC labelled the midbody remnant.

Another example is figure 4.

The authors make a claim that the spindle moves along the cortex and tilts, but only show one image, they should quantify that (for example to support their claim, they could show that the spindle shifts over a distance that is half the length of the spindle, thus allowing it to allow the extrusion of the polar body 2 just under polar body 1). In figure 5 and 6, the authors show that chromatin proximity is sufficient to induce outpocketing, similarly to mouse oocytes, but only show 2 images (one in the figure, one in supplementary). They should quantify it.

Please see added quantification in Figure 4 legend:

The meiotic spindle rotates from a parallel orientation by 50°+/- 3.1°, n=13, mean +/- SEM

  • The authors make the point that the midbody is responsible for the position of the outpocketing of the second polar body. (For example with sentences such as “Here in the ascidian we have discovered that during meiosis II the midbody formed at PB1 emission participates in directing the precise site of PB2 outpocketing”

This sentence has been changed to: Here in the ascidian we suggest that during meiosis II the midbody formed at PB1 emission predicts the precise site of PB2 outpocketing.

in the introduction or “we propose here in ascidians that the precise location of PB2 is dictated by the midbody remnant left behind in the egg following emission of PB1”

This sentence has been changed to:

…we speculate that the precise location of PB2 is dictated by the midbody remnant left behind in the egg following emission of PB1. We show that the midbody remnant formed during PB1 formation likely remains in the egg following PB1 emission and sits at the apex of the cortical outpocket that will form PB2.

As they state at the end of the manuscript, this is purely correlative. It could be that the midbody is retained at the site of maximum curvature (which could be quantified), that the midbody is retained by the actin cap which in turn controls where the outpocketing is… They should tone down their claim.

We have followed the advice of the reviewer and toned down the claim throughout the article.

Alternatively, they could try to demonstrate that the midbody is instructive to the position of the outpocketing. Is it not possible to do midbody preps such as Addi et al., 2020 and add midbodies in suspension to the media? This reviewer understands that it might be very difficult to get the midbodies out of the oocytes, but they could use separate oocytes, or embryos, or even adult tissues? Similarly, the statement in the discussion “our evidence indicates that actin is recruited around the midbody early during meiosis II” seems like an overclaim. It just seems like the midbody sits near the actin cap; the relationship between the two is not clear, and causality could only be demonstrated by moving the midbody (perhaps using a microneedle, or by treating the oocytes with a midbody solution).

Due to the current Covid pandemic we are currently locked out of the laboratory for all but essential animal maintenance. Further experiments are thus not possible.

The authors also suggest that the midbody could be responsible for spindle rotation and envision a variety of possible scenario. Again, they should tone down their claims as it is not clear whether the midbody has any influence on spindle positioning. Thus, the discussion is widely hypothetical, and it should be clear that it is.

The first paragraph of the disciussion has been altered to reflect the fact that this article is based on correlations, and as such is speculative:

Although a Ran-GTP gradient emanating from the chromosomes of the meiotic spindle causes the overlying cortex to polarize in readiness for PB formation (Dehapiot and Halet, 2013; Deng et al., 2007), we suggest that the precise location of PB2 may be dictated by the midbody remnant left behind in the egg following emission of PB1. This article is based on correlations and as such is highly speculative. The following discussion is therefore based on the possibility that midbody remnants influence the positioning of the second polar body. We suggest that the midbody remnant formed during PB1 formation likely remains in the egg following PB1 emission and sits at the apex of the cortical outpocket that will form PB2 

  • The evidence that the midbody is within the oocyte is not convincing. As it stands, the staining show that 1) there is cortical actin near the midbody 2) the midbody is closely associated with both PB and oocyte. Thus, it is not clear whether the midbody is release through abscission on both sides then recaptured by the oocyte, or only severed on the polar body side, or, more likely, not severed at all, explaining the tethering. As it is not clear exactly what happens here in terms of midbody release, the authors should either test what they are claiming (by providing evidence that the midbody is within the oocyte, for example by providing membrane staining, electron microscopy images, or live imaging of abscission on the PB side for example), or tone down their claim.
  • We have added the paragraph: These data do not formally indicate that the midbody remnant remains within the egg following removal of PB1 since there remains the possibility that the midbody remnant is stuck onto the egg surface. Also note that although the egg chromosomes are located near the midbody they are not always located directly under the midbody (Figure 3Biii). In addition, we have found recently that anti-phospho aPKC (atypical protein kinase C) strongly labels the midbody (Pruliere et al., in preparation). We thus also examined eggs that had PB1 removed with anti-phospho aPKC to label the midbody, anti-tubulin for microtubules and DAPI for DNA. The midbody remnant was again found to be located with the egg following removal of PB1 (Figure 3C). We thus speculate that the first midbody remains within the egg following PB1 emission.

We have also altered the discussion further, see paragraph:

  • Based on the results presented here, we speculate that the midbody directs the precise site where the actin cap forms during meiosis II, and that this cortical site prefigures the precise location of cortical outpocketing during PB2 emission;
  • Similarly, in the discussion, the authors state: “This situation is similar to the finding in somatic cells where midbody remnants remain in one of the two daughter cells (Chen et al., 2013; Guizetti and Gerlich, 2010) rather than being externalized following cell division (Crowell et al., 2014).” These papers do not really say that. Guizetti and Gerlich does not mention the fate of the midbody. Crowell describes what happens after the midbody gets released then recaptured by a cell.

We have replaced the Guizetti and Gerlich 2010  review with a more recent original article showing that the midbody remnant is retained by one daughter cell (Singh D, Pohl C  (2014). Dev; Cell Feb 10;28(3):253-67.) We thank the reviewer for spotting the lack of clarity, even though this is a review article we concede that it covers all types of midbody behavior.

However, we have chosen to retain one of the two reviews and would prefer to retain the more recent Chen et al review since it also more clearly states that:  These approaches showed that post-mitotic MBs associated with one of the two daughter cells after abscission [3, 18, 19, 21, 22].”

Finally, we cited the Crowel article by stating “…rather than being externalized following cell division…”. We cited this article correctly to indicate that midbodies are sometimes not retained by one of the dividing cell. We therefore intend to retain this citation for clarity and completeness.

  • The figures should be presented to be easier to read: for example, the stainings should be indicated on the panel and not only in the legends.

Minor comments:

the abstract should state that this study is in ascidians, it is not clear at first glance.

            Ascidian now appears in the abstract.

It is not clear what figure 3A-C brings to the study.

            3A has been removed.

species name should be in italic

            Corrected throughout

The methods should be readable without having to go back to a previous paper (avoid “as described previously”

            Methods changed as requested:

            For example, ascidian eggs were treated with 1% trypsin solution for 90min, gently            pipetted and the denuded eggs free of their extracellular chorion were washed three        times to replace the trypsin seawater with fresh seawater.

            Microinjection was performed by preparing a inserting about 50 eggs into a holding        chamber (wedge) made from glass pieces stuck to a 22mm coverslip with valab        (Vaseline:lanolin:beeswax 1:1:1)

Round 2

Reviewer 2 Report

By toning down their claims and adding quantifications, the authos have adequatly adressed my concerns.

Minor comment to be fixed before final publication: the table is not properly inserted in the file, it is messing up the line numbers.